# Retrospective Study of Maxillary Sinus Augmentation Using Demineralized Tooth Block Bone for Dental Implant

**DOI:** 10.3390/bioengineering11060633

**Published:** 2024-06-20

**Authors:** Hyunsuk Choi, Dong-Seok Sohn

**Affiliations:** 1Department of Dentistry and Prosthodontics, Daegu Catholic University School of Medicine, Daegu 42472, Republic of Korea; hschoi@cu.ac.kr; 2Department of Dentistry and Oral and Maxillofacial Surgery, Daegu Catholic University Medical Center, Daegu 42472, Republic of Korea

**Keywords:** sinus augmentation, tooth bone graft, dental implant

## Abstract

(1) Background: When placing implants in the maxillary posterior region with insufficient alveolar bone, a maxillary sinus elevation is necessary. Autogenous bone, though biologically ideal, poses risks and discomfort due to donor site harvesting. Block-type autogenous tooth bone graft material, made from the patient’s own extracted tooth, offers similar biological stability without these drawbacks. (2) Methods: This study observed the progress of 19 implant patients who were treated with maxillary sinus elevation procedures using block-type autogenous tooth bone graft material at the Daegu Catholic University Medical Center. Extracted teeth were processed into demineralized tooth block bone. After elevating the sinus membrane, implants and the tooth bone graft material were placed in the space, and the bony window was repositioned. Postoperative evaluations through clinical observation and radiographic imaging assessed sinus membrane elevation, alveolar bone height increase, and implant osseointegration. (3) Results: Results showed proportional increases in alveolar bone height to the graft material size, with long-term stability. No postoperative complications occurred, even with sinus membrane perforation, and implants remained stable. (4) Conclusions: The study concludes that maxillary sinus lifts using block-type autogenous tooth bone graft material provide excellent bone induction and biocompatibility, making this a highly beneficial method for both dentists and patients.

## 1. Introduction

The restoration of partially edentulous and edentulous areas using dental implants has been widely applied since Branemark discovered the phenomenon of osseointegration, and is now regarded as the most ideal treatment method for reconstructing lost teeth [1]. Although the success rate of implants has gradually increased with advancing technology, there are still instances of implant failure. Particularly, when placing implants in the posterior maxilla, challenging situations such as thin and low alveolar ridges, poor bone quality, or maxillary sinus pneumatization can occur [2,3,4]. Anatomical variants like the configuration of the maxillary sinus septum or the presence of alveolar antral artery especially may act as risk factors for implant placement [5].

For these reasons, removable dentures have long been used as a traditional and successful treatment option with predictable outcomes for patients who have lost their maxillary molars. In the case of removable dentures, natural teeth or implants in the anterior region are used as surveyed crowns, and the posterior extension area is covered by traditional removable partial dentures. These removable dentures can be provided at a low cost, do not require additional surgical procedures, and are reversible, making them a less burdensome option. However, they do have disadvantages in terms of patient satisfaction, such as lower masticatory efficiency and psychological comfort compared to fixed prostheses [6].

Thanks to recent advancements in implant technology, placing implants in the maxillary molar area has also become easier. Unsuccessful implant surgery in the molar part of the maxilla has been associated with unsatisfactory residual bone height, insufficient alveolar ridge volume, and disadvantageous bone quality [7,8,9]. These challenges have been addressed by increasing bone height through maxillary sinus elevation surgery [10,11,12]. A variety of maxillary sinus augmentation surgeries have been utilized to restore the maxillary posterior region in both simultaneous and delayed implant placement cases.

The lateral window sinus elevation technique is a commonly used surgical procedure, particularly when the height of the alveolar ridge is not high enough to achieve initial stability of the simultaneously placed implant [10,11,12]. In the case of the lateral approach technique, the sinus membrane (Schneiderian membrane) is elevated, and powdered bone graft materials are conventionally used to fill the new space beneath the elevated membrane [9]. A variety of powdered bone graft materials are utilized to fill the space. For example, autogenous bone, allogenous bone, xenogenous bone, synthetic bone, and various combination of these materials have been used [13,14,15].

However, sticking points which are related to graft materials still exist. For example, these include unexpected bone resorption, limited quantity of autogenous bone, donor site morbidity, lack of osteoinduction potential in allograft bone, and slow resorption speed of xenograft bone [16,17,18].

In addition, filling the maxillary sinus space requires a significant amount of bone grafting. In general, to fill one maxillary sinus, approximately 5–6 cc of autogenous bone or 2–3 cc of combination bone graft material is required. On the other hand, some studies have attempted to promote new bone formation by filling the maxillary sinus space with the patient’s own blood or platelet-rich plasma (PRP), without the use of bone graft materials [19,20,21].

Recently, there has been an increase in the use of demineralized autogenous tooth as a bone graft material in alveolar bone augmentation and maxillary sinus augmentation procedures, based on the realization that demineralized autogenous teeth can serve as excellent bone graft materials [17,22]. Demineralized autogenous tooth bone can be made from extracted teeth, including third molars and those extracted due to periodontal disease or dental caries. Previous studies have shown that human teeth, as biocompatible materials, exhibit both osteoinductive and osteoconductive characteristics [23]. However, demineralized autogenous tooth bone has the disadvantage of limited quantity for use in maxillary sinus augmentation. To address this issue, block-type tooth bone may offer advantages in terms of maintaining maxillary sinus space compared to powder-type tooth bone [24].

In this study, a retrospective analysis was conducted to evaluate the use of the lateral approach to sinus augmentation with autogenous demineralized tooth block bone in patients with residual bone height of 6 mm or less in the maxilla.

## 2. Materials and Methods

### 2.1. Patient Selection

This clinical retrospective study population comprised 19 patients from April 2017 to October 2022 and was approved by the local university ethics committee. Data were obtained from the dental treatment records of patients of the Daegu Catholic University Medical Center. All patients were notified about the process of treatment and signed consent forms for the surgery.

Before treating patients, each patient’s medical history was thoroughly examined, and the patients who had diseases affecting bone metabolism were ruled out. Smokers were notified that smoking could negatively impact the prognosis of sinus augmentation and decrease the implant success rate, but they were not ruled out from the study. All subjects had no contraindication for surgical procedures. The surgeries were carried out at the partial or complete edentulous premolar and molar part of the maxilla using the same surgical procedure.

The criteria for implant removal were determined when there was mobility in the implant fixture, which was considered as implant failure. The survival rate of the implants was estimated by recording the period from initial placement of the implant to the removal of the implant. If the implant was not removed, the date of the patient’s last visit for a routine checkup was used as the reference. Radiographic exams were conducted for preoperative analysis.

Panoramic radiographs and dental cone beam computed tomographic scans (Combi, Pointnix Co., Seoul, Republic of Korea and i-Cat, Imaging Sciences, Hatifield, PA, USA) were taken to estimate the condition of the maxillary sinus and the residual alveolar bone height (Figure 1).

With these software programs, we checked the residual bone height in millimeters. The width of the maxillary sinus was recorded based on the widest point of the sinus cavity. Because the elevated sinus membrane goes down when there is no tenting material, the amount of membrane elevation was estimated by measuring the distance from the base of the maxillary sinus floor to the top of the grafted tooth block bones.

### 2.2. Preparation of Osteoinductive Tooth Block Bone

A hopeless tooth was extracted carefully and processed into demineralized tooth block bone using the following technique. The attached soft tissue, decay, calculus, or composite restorations were removed with a #8 round carbide bur (Kerr, Orange, CA, USA). The tooth was divided into halves along its long axis with a disk while irrigating with sufficient coolant. The pulp attached to teeth was removed with a #4 round carbide bur (Kerr, Orange, CA, USA). To prevent dentin collagen denaturation by heat, all procedures were performed under irrigation with saline solution. 0.5 mm wide micro-holes were created at 5-6 mm intervals in each tooth block using a #2 round carbide bur (Kerr, Orange, CA, USA), allowing for the infiltration of blood into the tooth block bone (Figure 2).

Each perforated tooth block was placed into a vacuum ultrasonic device (VacuaSonic System, CosmoBioMedicare Co., Seoul, Republic of Korea). First, sterilization of the tooth blocks was conducted using a sterilization reagent (peracetic acid ethanol solution). After sterilization, the tooth blocks were demineralized for 60 min using 0.6 N hydrochloride under vacuum compression and ultrasonic vibration. Finally, the demineralized tooth blocks were sequentially rinsed with phosphate-buffered saline (PBS) and distilled water in the same device (Figure 3 and Figure 4).

### 2.3. Surgical Protocol

In this study, all patients were administered prophylactic antibiotics according to the same pharmacological protocol. Each patient was instructed to take oral antibiotics, i.e., 500 mg amoxicillin sodium (Augmentin, Ilsung pharmaceutical, Seoul, Republic of Korea), three times a day, starting one day before the surgery and continuing for seven days post-surgery. Prior to sinus grafting, 20–60 CC of venous blood was drawn from the patient’s forearm. The blood was collected in two non-coated vacutainers. After placing the vacutainers in a centrifuge machine (Medifuge, Silfradent srl, Sofia, Italy), they were centrifuged at 2400–2700 rpm for 2 min to obtain autologous fibrin glue, which was used to create sticky bone, as was carried out by Sohn et al. [7]. The upper layer shown on the non-coated vacutainer after centrifugation was mixed with biomaterials to make sticky bone grafts. While the non-coated vacutainers were centrifuged, the patient’s venous blood was collected in four to six glass-coated vacutainers and centrifuged for 12 min using the same centrifuge to make concentrated growth factor (CGF) fibrin blocks.

The surgery was conducted under local anesthesia using 2% lidocaine with 1:100,000 epinephrine. Lateral-approach sinus augmentation was planned for all patients. First, the mucoperiosteal flap was elevated, and the bone wall of the sinus was exposed. Then, a thin saw tip (S-Saw, Bukboo Dental Co., Daegu, Republic of Korea) was applied to the piezoelectric device (Surgybone^®^, Silfradent srl, Sofia, Italy) to create a bony window on the lateral wall of the maxillary sinus for access to the sinus mucosa and cavity.

An anterior vertical osteotomy line was designed 3 mm distal from the anterior vertical wall of the maxillary sinus. The superior osteotomy line was designed approximately 15 mm away from the anterior vertical osteotomy line. The height of the vertical osteotomy was about 10 mm.

The anterior and inferior osteotomy lines were angled at 45 ° to the lateral wall of the sinus. The superior and posterior osteotomy lines were made vertically to the lateral wall. With these four osteotomy lines, we created a trapezoidal-shaped bone window (Figure 5A). Through this piezoelectric osteotomy, the replaceable osteoinductive bony window (ROBW) could be preserved. This bony window served as a barrier over the fibrin-rich block inserted into the maxillary sinus (Figure 5A).

The bony window was detached by gentle levering action to reveal the sinus membrane. The sinus membrane was carefully separated from the sinus floor walls with a blunt, flat surgical instrument. Elevation of the sinus membrane was carried out until it reached the medial and posterior walls of the sinus cavity.

After the completion of sinus membrane elevation, the implant placement was performed immediately. In cases where the height of the residual bone was low or the bone quality was poor, the guide drill was used one step undersized compared to the original protocol to ensure the initial stability of the implant before placing the implant fixture. In the case of subject number 19, the insertion torque of each implant was 15 Ncm. Following the placement of implants, one or two tooth blocks were positioned over the implant apex (Figure 5B). Two to six pieces of fibrin-rich blocks containing concentrated growth factors (CGF) were inserted into the newly formed space beneath the tooth block bone to accelerate bone regeneration (Figure 5C). Then, the detached bony window was repositioned to its original location, preventing the in-growth of soft tissue into the sinus, and contributing to new bone formation (Figure 5D).

When there was insufficient alveolar bone width, a procedure for alveolar bone augmentation using tooth bone or other bone substitutes was performed (Figure 6A). After completing the bone graft, a collagen membrane was applied (Figure 6B). Then, flaps were sutured using polytetrafluoroethylene sutures (Cytoplast, Osteogenics Bio-medical, Lubbock, TX) with the continuous locking suture technique and the interrupt suture technique to achieve passive primary closure (Figure 6C,D).

Plain panoramic radiographs and cone beam computed tomograms were made immediately after surgery (Figure 7). Patients were informed to be cautious about blowing their noses, coughing, or sneezing forcefully for two weeks after maxillary sinus surgery. Stitching out of the sutures was performed 10 days after surgery. A healing period of 5 to 8 months was allowed for new bone consolidation and the osteointegration of implants. Before the second-stage implant surgery, standard panoramic radiographs and dental cone beam computed tomographic scans were performed (Figure 8).

The prosthetic procedure for the implants was conducted. After a provisional loading period of three months, the final full-zirconia restoration was delivered to the patient (Figure 9). A follow-up examination was conducted at an average of 53.2 months for all patients, and no periodontal and prosthetic concerns were observed in any of the patients (Figure 10).

## 3. Results

### 3.1. Clinical Results

Table 1 shows the clinical findings of this study. A total of 19 patients (11 male and 8 female) with a mean age of 53.9 agreed to be registered in this study, and no implants failed during this study period. Sinus augmentation was performed at both maxillary sinuses in 1 patient and on one side in 18 patients. A total of 20 sinus membrane elevation procedures were performed, and 27 implants were placed simultaneously.

Except for temporary swelling at the operation site, there were no serious complications in any of the patients during the follow-up period. There was no symptoms or signs of sinus infection or sinus-related disease either.

During sinus membrane elevation, sinus membrane perforation (less than 3 mm) occurred in three cases. Absorbable collagen (CollaTape; Zimmer Dental, Carlsbad, CA, USA) was used to manage these perforations. The overall survival rate of the implants was 100%. No implants were in a mobile state until the end of the study. Also, the sinus membrane perforation in three patients did not impact clinical outcomes. All patients underwent regular examinations for at least 2 years after final delivery of the implant prostheses, and there were no prosthetic or periodontal complications.

### 3.2. Radiographic Results

The radiographic results are tabulated in Table 2. The width of the maxillary sinus was measured at the widest point of the sinus cavity. Because the elevated sinus membrane goes down when there is no tenting material, the degree of membrane elevation was estimated by measuring the distance from the base of the maxillary sinus floor to the top of the grafted tooth block bones.

The preoperative bone height varied from 1.0 to 6.0 mm (SD 3.67 ± 1.49). The maxillary sinus width varied from 12 to 25 mm (SD 18.84 ± 4.15). Postoperative cone beam computed tomographic scans revealed that venous blood and voids filled the sinus space beneath the elevated sinus membrane. Until the end of the study, in all cases, infection signs in the maxillary sinus were not found in the radiographic images. Also, obvious increases in alveolar bone height and new bone formation with good continuity were found in the radiographic images. In all the cases, the formation of a new maxillary sinus floor around the apex of the implant was observed. There was no peri-implant marginal bone loss in any of the cases. There was no observed correlation between residual bone height and the increment in bone height. Also, no correlation between the maxillary sinus width and the increment of bone height were not found. But the amount of membrane elevation and new bone formation showed a positive correlation.

## 4. Discussion

When performing implant surgery in the thin maxillary alveolar ridge area, it is important to perform sinus augmentation and implantation at the same time. This is because, in this way, we can obtain a stable volume of the grafted sinus, minimize the operation time, shorten the total healing period, and minimize the surgery fee.

The lateral approach is a conventional and common method of sinus augmentation. This technique allows for sinus elevation through a lateral bony window. Using this technique, the sinus membrane can be directly viewed, allowing the surgeon to directly approach the sinus floor and detach the sinus membrane. Additionally, this technique is quite predictable and simple [17]. Generally, powdered graft materials from various bones are grafted into the elevated sinus. However, this technique requires a large amount of graft material, and if the graft materials penetrate the sinus membrane, it may cause maxillary sinusitis [19,25].

Some studies have reported successful sinus augmentation using PRP and peripheral blood without any bone grafting. Pinchasov et al. reviewed 19 scientific studies that researched new bone formation in sinus augmentation without using any bone graft material from 1993 to 2013 [26]. In this study, the preoperative alveolar bone height averaged over 5 mm, and the increase in bone height ranged from 4.5 to 8.2 mm. Also, space-maintaining devices were used to improve the increase in bone height. For example, hollow hydroxyapatite space maintainers and titanium bone fixation instruments were used. In addition, Gerardi et al. reviewed the efficacy of platelet-rich fibrin (PRF) in maxillary sinus lift procedures [27]. In this study, the regenerative capabilities of growth factors in PRF positively influenced new bone formation within the maxillary sinus.

Xu et al. reported that, in cases where sinus surgery was performed using only blood clots without bone grafts, most of the newly formed bone in the maxillary sinus had disappeared by 10 weeks [28]. The augmented bone height remarkably shortened from 2 to 10 weeks. This was because of the increased positive air pressure in the sinus cavity, which facilitated the pneumatization of the sinus. The authors proposed that the blood clots might not bear the positive air pressure.

Therefore, xenogenous bone graft material was used to prevent resorption of the grafted material caused by the positive air pressure. Until a certain period of time had passed after the surgery, the augmented height was actually stable [16,28]. These findings suggest that the xenograft particles endured air pressure, which induced osteoclastic bone resorption [28]. However, the xenogenous bone graft material seemed to be slowly resorbable or even non-resorbable for up to 6 years, as proven by clinical biopsies [29,30]. Non-resorbable bone does not change into new bone, but rather functionally adapts to the surrounding bone.

The tooth bone was chosen as an alternative to address the drawbacks of the other bone graft materials mentioned above. Tooth bone is autogenous graft material processed from a patient’s own extracted teeth, and thus, there are no complications at the donor site. We made the extracted teeth into autogenous demineralized tooth block bones.

Demineralized autogenous tooth ring and block bone grafts are beneficial solid space makers that overcomes several drawbacks of autogenous ring and block bone grafts, aiding in vertical augmentation of the extraction socket and ridge defect. The inorganic and organic components that constitute dentin are similar to those of alveolar bone [23]. Autogenous tooth bone has been used as a bone graft material regardless of whether it is demineralized or not. Tooth bone can be utilized in both powder and block forms [31,32,33]. When the autogenous undemineralized dentin matrix is used as a bone graft material, it shows favorable outcomes, with gradual replacement by newly formed bone. This indicates that tooth bone possesses significant structural and biological advantages, making it a suitable alternative to autogenous block bone [34]. However, the undemineralized dentin matrix has a disadvantage of lacking osteoinductivity in a non-osteogenic area [35]. Demineralization of the tooth block is known to have a significant impact on new bone formation at the recipient site [36]. Through demineralization of the teeth, an increase in the number of exposed dentinal tubules occurs, leading to the enlargement of dentinal tubules and exposure of osteoinductive proteins. In addition, the decreased crystallinity of dentin allows for replacement resorption [37,38,39]. When tooth bones are utilized in block or ring forms, demineralization and micro-perforation of the teeth bones are necessary. Demineralized and micro-perforated tooth blocks induce faster and greater bone formation than undemineralized and non-perforated tooth blocks [40,41]. In addition, Kim et al. reported that demineralized tooth blocks have acted as three-dimensional scaffolds promoting new bone formation. It has been observed that they integrate well with the alveolar bone, resulting in significant maintenance of graft volume and minimal bone loss [42].

Using this autogenous demineralized tooth block technique, a small amount of bone graft material was needed compared to the conventional techniques. If autogenous demineralized tooth bone was processed into a block form, a single molar was sufficient for one maxillary sinus surgery. Also, the surgical procedure became simple, because only tooth blocks needed to be inserted into the maxillary sinus. Because the tooth block bones were made from extracted teeth, this was more cost-effective than using ready-made bone graft material. In addition, our data show that the autogenous demineralized tooth block bone contributed to the maintenance of the space by bearing the sinus air pressure.

Ideal characteristics of graft material for sinus augmentation are biologic safety, osteoinduction capability, and space maintenance while preventing sinus pneumatization. Autogenous demineralized tooth block bone for sinus augmentation provides the space maintenance to prevent pneumatization. In addition, some studies have reported that autogenous demineralized teeth stimulate osteogenesis [17,23,43]. This is because demineralized teeth contain abundant growth factors, such as transforming growth factor-beta (TGF-beta), fibroblast growth factor (FGF), bone morphogenetic proteins (BMP), platelet-derived growth factor (PDGF), and epidermal growth factor (EGF) [22,44].

In many lateral approach cases, the lateral wall of the maxillary sinus is ground by a dental bur to create a lateral window. After completing the sinus lift procedure, the grinded bony window area is commonly covered with either resorbable or non-resorbable membranes. In this study, we made lateral windows using a precise osteotomy technique and detached them. After the bone graft procedure, we repositioned them. Through this technique, not only did we prevent soft tissue from growing into the grafted site within the maxillary sinus space, but we also promoted new bone formation within the maxillary sinus cavity. Moreover, the cost of the membranes and the operation time to stabilize the membrane were reduced [45,46].

For the detached bony window to be correctly repositioned to its original location, clear and precise osteotomy is crucial. This is made possible by a thin-blade saw tip, which is combined with a piezoelectric device. The precisely created osteotomy line protects the detached bony window from falling to the sinus cavity. The use of the piezoelectric device has many advantages. It has more favorable new bone formation speed than other osteotomy devices, direct visibility over the whole operation site, precise bone cutting, and soft tissue protection ability which prevents artery damage [45,46].

A limitation of this study is that histological and histomorphometric analysis of the maxillary sinus were not performed. In a future study, new bone formation and the histomorphometry of the maxillary sinus can be evaluated through patient biopsies. In addition, although 100% effective results were shown within the limited period of this 4-year study, complications should be evaluated through longer-term research.

## 5. Conclusions

Through clinical and radiographic results, this retrospective study shows that using only demineralized tooth block bones allowed for successful sinus bone grafting without inflammatory reactions, damage to the donor site, or the use of manufactured graft materials. This study clearly shows that using demineralized tooth block bone in sinus augmentation can elicit a favorable prognosis. Therefore, this technique has been shown to have advantages for both patients and clinicians.

## Figures and Tables

**Figure 1 bioengineering-11-00633-f001:**
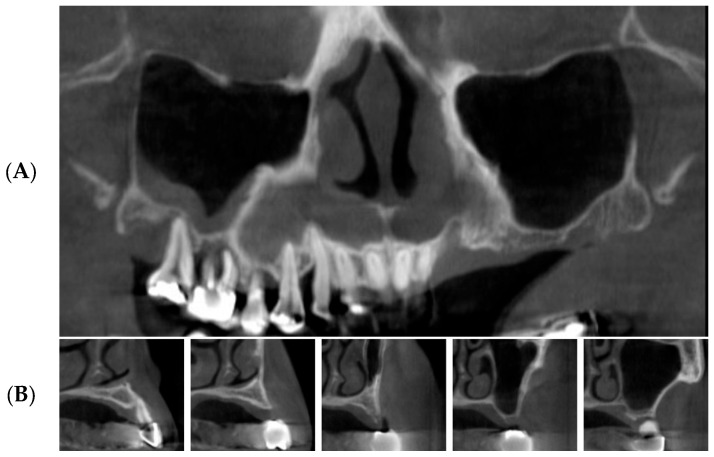
Preoperative panoramic CBCT images (subject number 19). (**A**) panoramic image of CBCT scan reveals unfavorable bone height on the upper posterior edentulous ridge due to pneumatization of left maxillary sinus. (**B**) cross-section images of CBCT show severe bone resorption.

**Figure 2 bioengineering-11-00633-f002:**
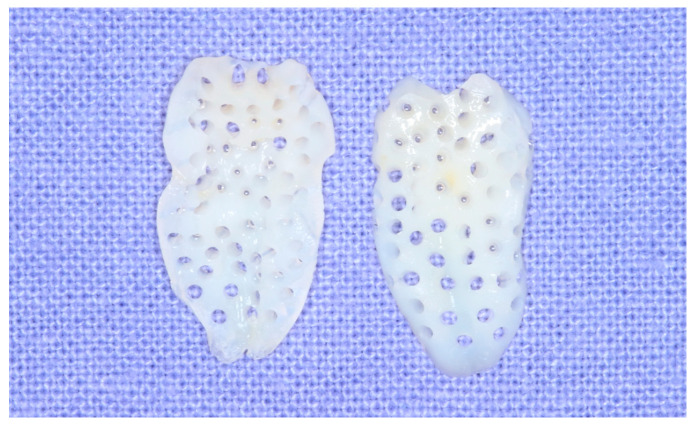
Demineralized and microperforated tooth blocks for maxillary sinus bone grafting.

**Figure 3 bioengineering-11-00633-f003:**
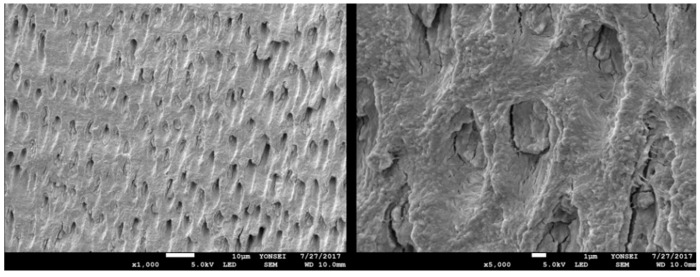
SEM (scanning electron microscope) images of undemineralized dentin show dentinal tubules blocked with hydroxyapatites. Consequently, diverse osteoinductive proteins are released with a delay.

**Figure 4 bioengineering-11-00633-f004:**
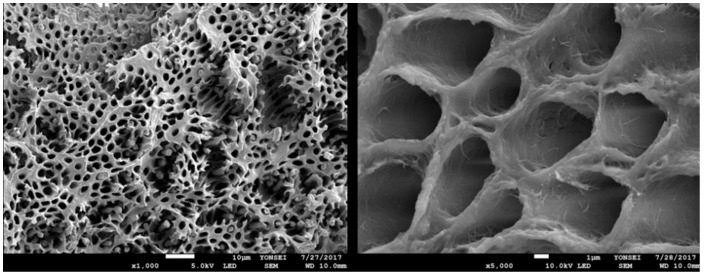
SEM images of demineralized dentin. Dentinal tubules are enlarged, allowing for rapid protein release, thereby promoting bone regeneration. Type I collagen fibers are also exposed, providing a scaffold for bone mineralization.

**Figure 5 bioengineering-11-00633-f005:**
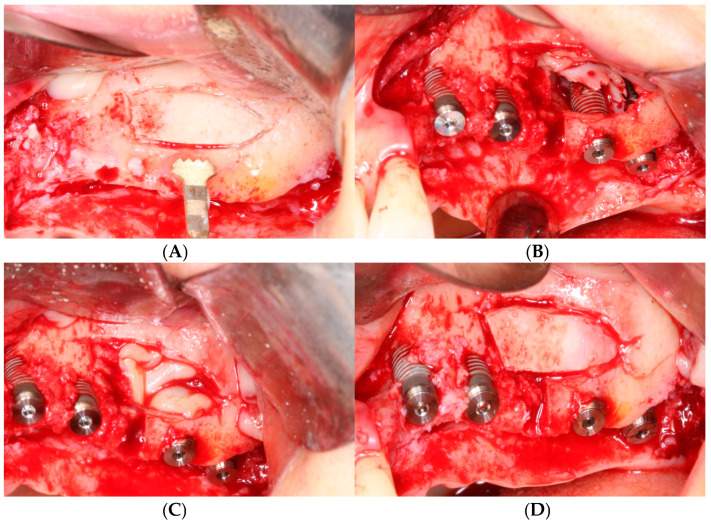
Surgical procedure of sinus augmentation using demineralized tooth block bones. (**A**) Replaceable osteoinductive bony window (ROBW) was prepared by using a piezoelectric device to expose and elevate the maxillary sinus mucosa. (**B**) Implants were placed with good stability after under-osteotomy with two tooth block bones placed over implant apex. Note severe horizontal bone deficiency around implants. (**C**) Fibrin-rich CGF blocks were placed in the newly formed sinus space. (**D**) ROBW was repositioned to its original location.

**Figure 6 bioengineering-11-00633-f006:**
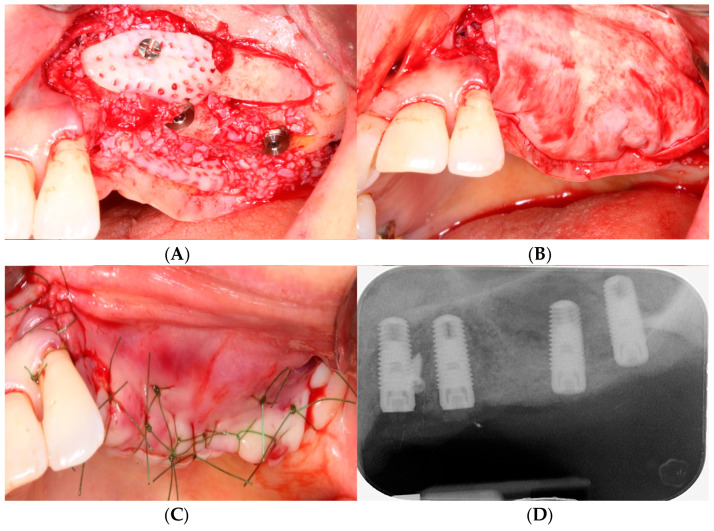
(**A**) After grafting sticky tooth bone powder onto the buccal and palatal defects of the alveolar bone, a tooth block was transplanted onto the buccal side and secured with a mini screw. Simultaneously, a tooth block bone was inserted beneath the palatal mucosa on the palatal side. (**B**) A collagen membrane was applied to the grafted bone. (**C**) Tension-free primary closure was achieved. (**D**) Postoperative periapical radiograph.

**Figure 7 bioengineering-11-00633-f007:**
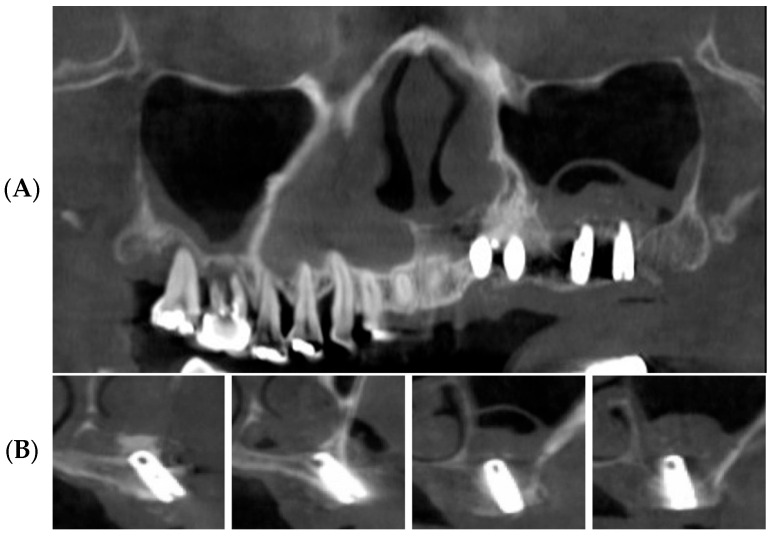
Postoperative panoramic CBCT images. (**A**) Panoramic image of CBCT scans reveals elevated sinus mucosa. (**B**) Cross-sectional images of CBCT scans reveals tooth blocks under the elevated sinus mucosa.

**Figure 8 bioengineering-11-00633-f008:**
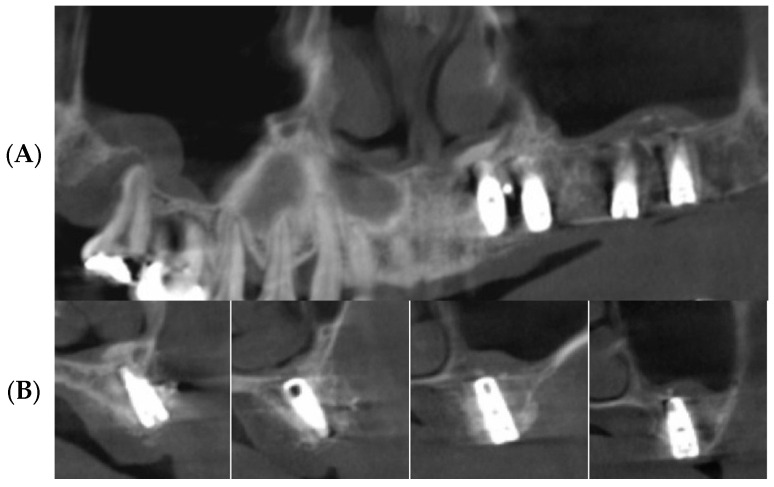
CBCT scans after 5 months of healing. (**A**) Panoramic image shows augmented maxillary sinus. (**B**) Cross-sectional images shows favorably augmented alveolar ridge and maxillary sinus.

**Figure 9 bioengineering-11-00633-f009:**
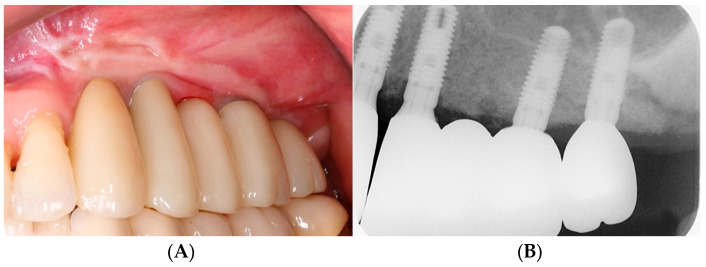
Delivery of final implant zirconia prosthesis. (**A**) Intraoral photograph. (**B**) Periapical radiograph.

**Figure 10 bioengineering-11-00633-f010:**
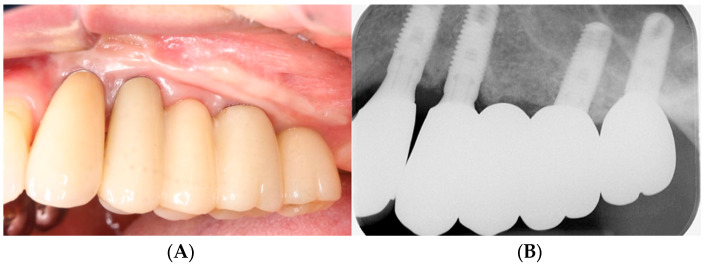
Follow-up examination after 5 years. (**A**) Intraoral photograph. (**B**) Periapical radiograph.

**Table 1 bioengineering-11-00633-t001:** Patient characteristics and clinical findings.

Subject	Sex/Age	Region	Failed Site	Sinus Membrane Perforation	Months to Second-Stage Surgery	Months in Function
1	M/50	#16	-	No	4	58
2	M/46	#26	-	No	6	42
3	F/50	#27	-	Yes	4.5	60
4	M/46	#14,15,16,17	-	No	7	51
5	M/59	#26,27	-	No	5	50
6	F/57	#26	-	No	5	48
7	F/52	#17	-	Yes	10	50
8	F/52	#27	-	No	5.5	55
9	M/65	#16	-	Yes	6.5	54
10	M/50	#16,26	-	No	7.5	59
11	F/58	#26,27	-	No	7.5	52
12	F/47	#27	-	No	4	58
13	M/64	#16	-	No	3	50
14	M/75	#26	-	No	6	60
15	F/54	#16	-	No	10	46
16	M/45	#17	-	No	3.5	49
17	M/52	#26,27	-	No	5	53
18	M/45	#27	-	No	10.5	54
19	F/57	#26,27	-	No	5	61

**Table 2 bioengineering-11-00633-t002:** Radiographic assessment of maxillary sinus and alveolar crest.

Subject	Region	Preoperative Bone Height (mm)	Maxillary Sinus Width (mm)	Amount of Sinus Membrane Elevation (mm)	Bone Height at the End of the Study (mm)	Increment in Bone Height (mm)
1	#16	5	12	18	14	9
2	#26	3	18	17	13	10
3	#27	5	16	16	14	9
4	#14,15,16,17	5,3,5,5	22	15,18,14,16	15,16,14,14	10,13,9,8
5	#26,27	4,4	16	16,16	15,12	11,8
6	#26	6	16	18	20	14
7	#17	2	24	16	15	13
8	#27	2	22	19	18	16
9	#16	2	22	16	14	12
10	#16,26	3,3	23	23,20	20,22	17,19
11	#26,27	3,2	24	19,18	20,16	17,14
12	#27	4	22	16	15	11
13	#16	5	12	14	14	9
14	#26	4	16	12	12	8
15	#16	1	20	22	18	17
16	#17	5	13	17	20	15
17	#26,27	6,6	15	10,11	15,16	9,10
18	#27	3	25	12	14	11
19	#26,27	1,2	20	21	14,14	13,12

## Data Availability

Data are included in the article.

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
