# Peer review of "Retrospective Study of Maxillary Sinus Augmentation Using Demineralized Tooth Block Bone for Dental Implant"

_bioengineering, 2024, doi:10.3390/bioengineering11060633_

Round 1

Reviewer 1 Report

Comments and Suggestions for Authors

Thanks for inviting me to review this article submitted to Bioengineering, which was about dental implant. The topic fell within the scope of the journal, but there were some important issues pending addressed. Please consider the following comments, and make a Resubmission.

1: According to the iThenticate system, the total similarity index of the submission reached 38%, which was a very high value of an article. Please carefully check and revise your text to avoid problems about publication ethics.

2: The reviewer would express the concern on Figure 9 and 10. Although they might be the photographs of the same patient before and after 5 years, the similarity was seemingly high. It looked like changing the photographing angle. Of course, some coincidence might exist and should not be judged by human eyes. The authors were advised to double-check the data.

3: Please clarify the information of the used dental materials.

4: As stated in the end of Introduction, “…with a residual bone height of less than 6 mm”. But in Table 2, subject #6 and #17 possessed residual bone height = 6 nm. Please reconsider the standard.

5: The format of References should be double-checked.

Author Response

1: According to the iThenticate system, the total similarity index of the submission reached 38%, which was a very high value of an article. Please carefully check and revise your text to avoid problems about publication ethics.

-> Thank you for your comment. We modified the paper and lowered the similarity level to 19%.

2: The reviewer would express the concern on Figure 9 and 10. Although they might be the photographs of the same patient before and after 5 years, the similarity was seemingly high. It looked like changing the photographing angle. Of course, some coincidence might exist and should not be judged by human eyes. The authors were advised to double-check the data.

-> Thank you for your thoughtful review. At our university medical center, clinical and radiographic photographs are taken at every follow-up. As a result of double-checking, the records are correct for 2017 and 2022, respectively. Please compare the changes in gingiva at the margin of the prosthesis and the healing pattern of scar tissue caused by the surgical incision line.

3: Please clarify the information of the used dental materials.

-> Thank you for your thoughtful review and considerations. We added the information of the used dental materials on our text.

4: As stated in the end of Introduction, “…with a residual bone height of less than 6 mm”. But in Table 2, subject #6 and #17 possessed residual bone height = 6 nm. Please reconsider the standard.

-> Thank you for your thoughtful consideration. We changed the text from “less than 6 mm” to “6 mm or less”.

5: The format of References should be double-checked.

-> Thank you for your comment. We unified the format of references.

Reviewer 2 Report

Comments and Suggestions for Authors

Dear authors, congratulations for your study and your contribution to the research. The structure of this study is well-organized and the investigation is comprehensive. However, I suggest some major revisions in order to improve the quality of the manuscript.

Introduction:

Before introducing the different types of rehabilitation and making a comparison between the removable dentures and dental implants, it would be interesting to provide a classification of the atrophies of maxillary bone.

Besides, it could be useful to introduce the anatomy of maxillary sinus and the sticking points related to two important anatomical variants which could affect the maxillary sinus augmentation procedures, such as the Underwood septa and the alveolar antral artery: I suggest to add the recent literature about the anatomy of maxillary sinus and the related anatomical variants; the following study is an example of the above described topic: https://doi.org/10.3390/tomography10040034

Materials and methods:

2.1 Patients selection:

I suggest to better highlight the inclusion and exclusion criteria; moreover, it would be useful to add different subparagraphs, such as “Inclusion and exclusion criteria”, “Radiographic evaluation”, “Clinical evaluation” etc..

Discussion:

Line 285: before listing the advantages of the lateral approach technique, it would be useful to deepen the technical description of it.

Line 290: Regarding the use of autologous platelet concentrates, authors cited some literature about the use of PRP which represents the first generation of autologous platelet concentrates: I suggest to update this topic introducing the second generation of PRF and the recent studies or review about its effectiveness in maxillary sinus augmentation procedures; the following review could be helpful https://doi.org/10.23812/j.biol.regul.homeost.agents.20233705.232

Line 351-365: it would be useful to cite some studies regarding the described bony window technique, the importance of performing a precise osteotomy and the use of piezoelectric, respectively.

Finally, I suggest to describe the limits of the study and the eventual limits of the proposed protocol.

Author Response

Introduction:

Before introducing the different types of rehabilitation and making a comparison between the removable dentures and dental implants, it would be interesting to provide a classification of the atrophies of maxillary bone.

Besides, it could be useful to introduce the anatomy of maxillary sinus and the sticking points related to two important anatomical variants which could affect the maxillary sinus augmentation procedures, such as the Underwood septa and the alveolar antral artery: I suggest to add the recent literature about the anatomy of maxillary sinus and the related anatomical variants; the following study is an example of the above described topic: https://doi.org/10.3390/tomography10040034

-> Thank you for your thoughtful consideration. We have added the content about anatomical variants of maxillary sinus to the introduction of our paper and also added the reference mentioned above.

Materials and methods:

2.1 Patients selection:

I suggest to better highlight the inclusion and exclusion criteria; moreover, it would be useful to add different subparagraphs, such as “Inclusion and exclusion criteria”, “Radiographic evaluation”, “Clinical evaluation” etc..

-> Thank you for your thoughtful review and considerations. We attempted to exclude from the study patients who were taking drugs or receiving injections that could affect bone metabolism, but no patients were excluded within this study period. Attention was also paid to smokers, but they were not excluded from the study. Therefore, the inclusion and exclusion criteria were not categorized separately. Also, in Materials and Methods, contents and figures are sequentially described so that the surgical procedure can be easily understood. In Results, clinical evaluation and radiographic evaluation are presented as subparagraphs, separately. We would appreciate your understanding in this regard.

Discussion:

Line 285: before listing the advantages of the lateral approach technique, it would be useful to deepen the technical description of it.

-> Thank you for your kind comment. We added the technical description about lateral approach technique to our text.

Line 290: Regarding the use of autologous platelet concentrates, authors cited some literature about the use of PRP which represents the first generation of autologous platelet concentrates: I suggest to update this topic introducing the second generation of PRF and the recent studies or review about its effectiveness in maxillary sinus augmentation procedures; the following review could be helpful https://doi.org/10.23812/j.biol.regul.homeost.agents.20233705.232

-> Thank you for your helpful comment. We have improved the content by adding the above journal to the discussion.

Line 351-365: it would be useful to cite some studies regarding the described bony window technique, the importance of performing a precise osteotomy and the use of piezoelectric, respectively.

-> Thank you for your consideration. We cited our studies regarding bony window technique and piezoelectric surgery on our text.

Finally, I suggest to describe the limits of the study and the eventual limits of the proposed protocol.

-> Thank you for your thoughtful consideration. We added the limitations of this study in the discussion.

Reviewer 3 Report

Comments and Suggestions for Authors

The topic of the reviewed article is “Retrospective study of maxillary sinus augmentation using demineralized tooth block bone for dental implant.
The aim of the study was to present the results of a retrospective study on the use of blocks from demineralized teeth as a graft material in the augmentation of the maxillary sinus and preparation for implant placement.
The authors describe a method of using the patient's previously extracted teeth as bone substitute material.
In the introduction, the authors clearly present the issue of implantation in the lateral part of the maxilla, where there is often insufficient bone volume.
The authors conducted research on 19 patients between April 2017 and October 2022.
The authors clearly present the criteria for qualification for the procedure and the methods of radiological assessment both before and after the procedure.
The authors presented in detail the pre- and post-treatment procedure and synthetically described the results they obtained.
The description of the procedure for processing the tooth into an augmentation material, which is the core of this manuscript, is clear and legible, which positively affects the reader's reception of the article.
A total of 20 sinus lift procedures were performed, introducing 27 implants. In 3 cases, the mucosa of the maxillary sinus was perforated.
The authors report that no implants were lost by the end of the study and that the average increase in bone height ranged from 4.5 to 8.2 mm.
Patients were subjected to systematic radiological control.
The article contains very helpful intra-procedure photos showing the technique and post-treatment effects, but the number of cases presented could be larger, especially considering such a long period of research.
The tables included in the article clearly present both pre- and post-treatment measurements.
In terms of grammar and style, I have no major objections.
The article, although based only on 19 cases, shows very optimistic data.
A follow-up period of approximately 4 years may not be sufficient to capture long-term results.

The results of this study suggest that the use of demineralized tooth blocks is reliable and 100% effective and does not carry any complications even in the case of rupture of the sinus membrane. As a practicing physician with many years of experience, I am always very afraid of surgical techniques that are 100% effective!
The authors did not present any complications after the technique used. I believe that even if the authors did not observe any complications in their material, they should familiarize the reader with possible complications by pointing out the experiences of other authors.
To sum up, the article is generally well written, in quite simple and understandable language, it provides valuable information about this specific treatment and its effects. It is a valuable source of information for practicing surgeons who can expand their skills with this technique in their own practice.

However, there are several aspects of this work that the authors should address and clarify in the text the comments indicated below:
-How long does the entire procedure take? Is it one or two stages?
-In the materials and methods in the "Patient selection" subsection there should be 19 patients, but there are 18. Why is this difference?
-What was the stabilization of the implants shown in the photo - partially covered with bone? It is advisable to provide the Ncm value at which the implants were inserted.
- Are all teeth eligible for the described procedure? Those with caries or also those retained? What are the criteria for qualifying a tooth for processing into an augmentation material?
- How long can such a tooth be stored after extraction and demineralization and under what conditions?
- Can demineralization be performed on an outpatient basis?
- How many blocks can a tooth be divided into and what thickness should these individual slices be?
- Were the perforations in the maxillary sinus membrane treated in any way?
- What is the loss of bone augmented with this method over time?
- Did the authors use the patient's own teeth as crushed biomaterial for transplants?  If so, what were the results?
- Do the authors have the consent of the bioethics committee for the research conducted? If so, please indicate the consent number and date. In particular, it should be clarified whether the radiological check-ups in patients had the character of medical observation of healing progress or whether they served as material for scientific research?

Author Response

-How long does the entire procedure take? Is it one or two stages?

-> Thank you for your comment. All patients underwent 2-stage surgery. Table 1 records the months to second stage surgery for each patient. As indicated in the text, patients typically underwent 3 months of temporary prosthetics before final prosthetic treatment. The entire procedure took approximately 10 months.

-In the materials and methods in the "Patient selection" subsection there should be 19 patients, but there are 18. Why is this difference?

-> Thank you for your thoughtful consideration. The text incorrectly states that there are 18 patients. We have corrected that error.

-What was the stabilization of the implants shown in the photo - partially covered with bone? It is advisable to provide the Ncm value at which the implants were inserted.

-> Thank you for your thoughtful review. The insertion torques of implants was 15Ncm, respectively. We have added this sentence to the article.

- Are all teeth eligible for the described procedure? Those with caries or also those retained? What are the criteria for qualifying a tooth for processing into an augmentation material?

-> Thank you for your consideration. Most teeth are eligible. If there is caries, calculus, or composite restoration, the tooth can be used after removal using a round carbide bur. There are no special qualifying criteria, but teeth with severe dental caries or posts or other restorations with very little usable tooth portion are excluded.

- How long can such a tooth be stored after extraction and demineralization and under what conditions?

-> Thank you for your question. Extracted teeth can be stored in the freezer for 6 months after sterilization, and if they need to be stored longer, they must be re-sterilized. After demineralizing process, it can be stored in the freezer for 2 months.

- Can demineralization be performed on an outpatient basis?

-> Thank you for your comment. Our hospital has an in-office dental laboratory, so teeth demineralization is possible using vacuasonic equipment (VacuaSonic System, CosmoBioMedicare Co., Seoul, Korea).

- How many blocks can a tooth be divided into and what thickness should these individual slices be?

-> Thank you for your consideration. In the case of tooth block bone, it can be processed into various sizes depending on the size of the tooth or the location where the tooth bone will be applied. At our hospital, tooth block bone can be processed into ring type, laminate type, and powder type. In the case of this study, two slices of 1 mm thick laminate block bone were made, and the rest were processed into powder type.

- Were the perforations in the maxillary sinus membrane treated in any way?

-> Thank you for your comment. Because only tooth block bone was used in our study, it is relatively free from the risk of postoperative infection or sinusitis compared to the case of using xenogeneic bone. When sinus perforation occurred, we completed sinus elevation, applied CGF fibrin blocks, and completed implant placement.

- What is the loss of bone augmented with this method over time?

-> Thank you for your question. Within the study period of this paper, bone loss was not observed in areas where maxillary sinus augmentation was performed. A longer-term follow-up study on this will be needed.

- Did the authors use the patient's own teeth as crushed biomaterial for transplants?  If so, what were the results?

-> Thank you for your comment. Currently in Republic of Korea, only one's own teeth are used as crushed biomaterial for reasons such as preventing immune rejection or infectious diseases. As can be seen from the results of this paper, the results were excellent.

- Do the authors have the consent of the bioethics committee for the research conducted? If so, please indicate the consent number and date. In particular, it should be clarified whether the radiological check-ups in patients had the character of medical observation of healing progress or whether they served as material for scientific research?

-> Thank you for your comment. This clinical retrospective study was conducted in accordance with the Declaration of Helsinki, and approved by the Institutional Review Board of Daegu Catholic University Medical Center Institutional Review Board (Approval IRB No. CR-22-174-L). Basically, radiological examinations were performed to evaluate the need for peri-implant periodontal treatment.

Round 2

Reviewer 1 Report

Comments and Suggestions for Authors

Thanks for your revision. The current version is acceptable.

Reviewer 2 Report

Comments and Suggestions for Authors

manuscript can be now accepted

Reviewer 3 Report

Comments and Suggestions for Authors

The authors approached the comments and remarks from round 1 with commitment. They explained all aspects that were important from the reader's point of view. In its current form, the manuscript is suitable for further processing towards publication.